# Refitting of Zirconia Toughening into Open-Cellular Alumina Foams by Infiltration with Zirconyl Nitrate

**DOI:** 10.3390/ma12121886

**Published:** 2019-06-12

**Authors:** Ulf Betke, Marcel Scheunemann, Michael Scheffler

**Affiliations:** Institute for Materials and Joining Technology—Nonmetallic Inorganic Materials and Composites, Otto-von-Guericke-University Magdeburg, Große Steinernetischstraße 6, 39104 Magdeburg, Germany; marcel.scheunemann@st.ovgu.de (M.S.); m.scheffler@ovgu.de (M.S.)

**Keywords:** cellular ceramic, ceramic foam, reticulated porous ceramic (RPC), transformation toughening, zirconia-toughened alumina (ZTA), hollow strut infiltration, metal salt infiltration

## Abstract

The present work describes the combination of the well-established dispersion infiltration of the hollow struts in reticulated porous ceramics (RPCs) and the salt solution infiltration of the remaining strut porosity. This approach is applied on alumina foams, which are loaded subsequently with a dispersion of sub-micrometer alumina particles and a ZrO(NO_3_)_2_ solution. The zirconyl nitrate is converted into a ZrO_2_ transformation toughening phase during the final sintering step. As a consequence of the complex microstructure evolution during the consecutive infiltration cycles, the reinforcement phase concentrates selectively at the weak spots of RPC structures—namely, the hollow strut cavities and longitudinal cracks along the struts. As a consequence, a severe improvement of the compressive strength is observed: The average compressive strength, normalized to a porosity of 91.6 vol.%, is 1.47 MPa for the Al_2_O_3_/ZrO_2_ infiltrated foams, which is an improvement by 40% with respect to alumina-only loaded foams (1.05 MPa) or by 206% compared to uninfiltrated alumina RPCs (0.48 MPa). The compressive strength results are correlated to infiltration parameters and the properties of the infiltration fluids, for example the rheological behavior and the size of the Zr solute species in the respective ZrO(NO_3_)_2_ solution.

## 1. Introduction

Open-celled ceramic foams are used within several technological fields; with respect to the quantity, the most prominent applications are filter materials for metal melts in casting or catalyst supports [1]. The production quantity of ceramic foam filters amounts to approximately 10^8^ pieces per year [2]. Ceramic foams for the above mentioned applications are manufactured on an industrial scale by the polymer sponge replication process established by Schwartzwalder and Somers in 1961 [3]. This technique is based on coating a ceramic dispersion on an open-cellular polymer foam template, a subsequent thermal removal of the polymeric support structure, and sintering for particle consolidation. After sintering of the green body, a ceramic replica of the initial foam structure is the result, which is similar to that of the (reticulated) polymeric sponge template. Consequently, these cellular structures are commonly addressed as “reticulated porous ceramics” (RPCs). A variety of cellular ceramic materials has been produced by the Schwartzwalder technique, an overview is given by Studart, et al. and Fey, et al., see refs. [4,5]. Novel aspects in the field of cellular ceramics include novel applications as well as novel processing and manufacturing routes. Recent examples for new applications are reactive filters in metal filtration as described by Storti et al., see ref. [6], RPCs as support structures in adsorption-driven heat transformation devices as described by Betke et al. in ref. [7], or syntactic ceramic foams as demonstrated by Rugele et al., see ref. [8]. With respect to the manufacturing processes, additive techniques like direct ink writing, inkjet printing in powder beds, or stereolithography are a focus of current research [9,10,11]. These additive manufacturing techniques add a severe amount of freedom with respect to the design of cellular structures. This goes far beyond stochastically irregular foams or regular honeycombs and allows the manufacturing of cellular structures with tailored properties [12,13]. However, up to now the quantities of 10^8^ cellular ceramics necessary per year in foundry applications, for example, cannot be handled by additive manufacturing. Thus, improvements to the processing and properties of conventional reticulated porous ceramics are still of current interest and represent the motivation for this study.

Intrinsically, RPC structures are characterized by a hierarchical pore system [14]: The actual cell pores with a size between 0.6 mm and 4.5 mm are visible to the naked eye representing the actual foam structure; however, due to the strut templating approach of the replica process, these cellular ceramics contain a characteristic triangular “hollow strut” cavity (d ≈ 100 µm) originating from the thermal template removal [7]. A third class of pores is found within the ceramic strut material itself, which can be adjusted mainly by the sintering conditions. This sub-micrometer porosity is suitable for the capillary force-driven loading with active materials or precursors, for example, metals salts, which may subsequently converted chemically into the corresponding metal species [15].

As the cross-section of the polymer foam struts is usually triangular and contains sharp edges, the application of a uniform coating with the ceramic dispersion is not trivial. Consequently, this results in a thinned-out coating on these sharp edges of the template structure, the so-called “Kantenfluchtproblem” (also known as blue edges) [16]. In many cases this is the origin of the typical flaws in RPC structures: longitudinal cracks running along the strut. This effect is especially perceptible for foams with a high total porosity; the dispersion coating thickness on the struts of the template foam is lower in this case. This results in a more pronounced thinning-out of the dispersion on the strut edges and consequently leads to a higher concentration of strut flaws [17]. In combination with the edges of the triangular hollow strut cavities acting as stress concentrators, these defects significantly reduce the overall mechanical strength of ceramic replica foams compared to cellular structures with dense struts made by other (foaming) techniques [18,19]. Nevertheless, these RPC-specific flaws open up a variety of modifications, for example, a functionalization of the inner strut regions. For these approaches the strut cracks are of essential importance as they connect the hollow strut cavities with the foam cells and allow the penetration of the strut interior [5].

Several improvements have been developed to mend and/or exploit these RPC-specific flaws, of which two are of general importance: (1) recoating, i.e., the application of a second ceramic dispersion coating on the green body RPC and (2) hollow strut infiltration, i.e., the filling of the hollow strut cavities in (pre-)sintered RPCs with a ceramic dispersion [5]. The recoating technique is experimentally simple as it can be applied on the dried green body without an additional sintering step using different methods [20]. However, the porosity and the effective cell size are reduced significantly with every recoating cycle [21,22,23,24], which tampers the fluid transportation properties of these structures. This problem is circumvented for the hollow strut infiltration, where cavities inside the strut material are filled preferably without a significant impact on porosity and thickening of the ceramic struts. The infiltration can be performed with a ceramic dispersion of the same raw material as the struts are made of, as well as a different ceramic phase [14,25,26,27,28,29]. Furthermore, the infiltration process can be performed at elevated temperature, either by using a melt or a reactive infiltration technique, and has been essentially applied in the manufacturing of Si-SiC foams for high temperature applications [30,31,32]. Nevertheless, the hollow strut infiltration requires the removal of the polymer template and a presinterpresintering step in order to stabilize the RPC structure prior to the further processing. Thus, this technique is experimentally more complex. Finally, with respect to an increase of the mechanical strength of RPCs, both approaches, infiltration and recoating, are effective with an absolute compressive strength increase of up to 400% [5].

A different approach towards mechanically stable (cellular) ceramics are transformation toughened ceramics, for example the well-known Al_2_O_3_/ZrO_2_ ceramic composites, or zirconia-toughened alumina (ZTA) [33]. The stress-induced phase transformation from tetragonal ZrO_2_, as retained in the alumina matrix, into monoclinic zirconia involving a significant volume increase of the ZrO_2_ grains is the reason for the strength increase observed for these composites [34,35]. The crack expansion is hindered by both, absorption of the activation energy for the *t*/*m* transformation of ZrO_2_, and the compressive stress exerted on the alumina matrix by the expanding ZrO_2_ grains [36]. Nevertheless, the application of this concept of transformation toughening into cellular ceramics is quite scarce, but effective in increasing the mechanical strength of porous and cellular ceramic materials [37,38].

In the present work, the concept of the hollow strut infiltration in alumina RPCs is combined with the transformation toughening through ZrO_2_ grains within the Al_2_O_3_ matrix. However, in contrast to previous studies [26,28], the zirconia phase is not provided as a component in the ceramic raw material. Instead, it is loaded into the strut material via capillary force-driven infiltration of the porosity within the alumina-infiltrated ceramic strut material using a zirconia precursor solution. As a consequence of the unique microstructure development of the alumina-infiltrated struts this allows a selective generation of the ZrO_2_ transformation toughening phase in the intrinsically weak regions of RPCs, namely the hollow strut cavities and longitudinal strut cracks. The focus of the work is set on the resulting compressive strength characteristics of the Al_2_O_3_/ZrO_2_ infiltrated alumina RPCs, which are correlated to the obtained strut microstructure and the infiltration conditions.

## 2. Materials and Methods

### 2.1. Sample Preparation 

The manufacturing process of ZTA ceramic foams described in this work can be subdivided in three main steps as illustrated in Figure 1: (a) manufacturing of alumina foams by the polymer sponge replication technique, (b) infiltration of the hollow strut cavities in these foams with alumina, and (c) zirconia loading by infiltration of the microstructural porosity with a zirconyl nitrate solution and a subsequent heat treatment step. For each sample series 15 specimens were prepared.

The initial open-cellular alumina foams were prepared according to the polymer sponge replication technique established by Schwartzwalder and Somers in 1961 [3]. Polyester polyurethane (PU) foams with the dimensions of 20 mm × 20 mm × 20 mm and a cell count of 20 pores per linear inch (S30P20R, Koepp-Schaum GmbH, Oestrich-Winkel, Germany) were used as a template structure. The PU template foams were coated with an aqueous alumina dispersion (Table 1) containing 78.3 wt.%, which is 47.8 vol.%, alumina (CT 3000 SG, Almatis GmbH, Ludwigshafen, Germany), 0.8 wt.% ethanolammonium citrate deflocculant (Dolapix CE64, Zschimmer & Schwarz Chemie GmbH, Lahnstein, Germany), 1.2 wt.% polyvinylalcohol binder (Optapix PA 4G, Zschimmer & Schwarz GmbH, Lahnstein, Germany), and 0.1 wt.% polyalkylene glycolether defoamer (Contraspum K1012, Zschimmer & Schwarz GmbH). The excess dispersion was removed from the foam cells by careful manual compression of the template. The weight of the coated PU foam was adjusted to 2.1 ± 0.1 g resulting in a final porosity of 93.8 ± 0.3 vol.% after drying, thermal template removal and sintering of the green body at 1350 °C. Details about the alumina dispersion formulation and the alumina foam manufacturing procedure are described elsewhere [39,40].

The subsequent infiltration of the hollow strut cavities with alumina was performed using an Al_2_O_3_ dispersion (Table 1) containing 63.8 wt.%, which is 30.9 vol.%, of sub-micrometer alumina powder (Taimicron TM-DAR, Taimei Chemicals Co., Ltd., Tokyo, Japan), 0.6 wt.% deflocculant (Dolapix CE64), and 0.1 wt.% defoamer (Contraspum K1012). The foams were completely immersed into the dispersion at ambient temperature and pressure under occasional, careful agitation. After 10 min the foams were collected from the dispersion, placed inclined on a smooth surface for 20 min, and periodically turned in order to allow the excess dispersion to rinse off of the foam cells. Finally, the hollow strut-infiltrated foams were dried under ambient conditions.

The infiltration with zirconyl nitrate was performed by placing the dried, strut-infiltrated alumina foams into a 30 wt.% solution of zirconyl nitrate hydrate (Sigma-Aldrich Chemie GmbH, Taufkirchen, Germany) in H_2_O or in HNO_3_ (*c* = 1 mol∙L^−1^ or 2 mol∙L^−1^), respectively. For one sample series, the ZrO(NO_3_)_2_ was doped with 3 mol% Y(NO_3_)_3_∙6H_2_O (ABCR, Karlsruhe, Germany) in 2 mol∙L^−1^ HNO_3_ as solvent. The foams were completely immersed into the solution and left there for 1 h. Afterwards, the foams were withdrawn and after the removal of excess zirconyl nitrate solution by careful centrifugation (60 min^−1^, 10 s) the samples were dried under ambient conditions. Subsequently, the ZrO(NO_3_)_2_ loaded foams were calcined in air at 600 °C for 2 h to decompose the zirconyl nitrate into ZrO_2_. For selected samples two or three consecutive cycles of ZrO(NO_3_)_2_ infiltration and calcination steps were performed. Finally the resulting Al_2_O_3_-ZrO_2_ foams were sintered for 3 h in air at 1650 °C.

As a reference sample series, strut-infiltrated alumina foams were treated with an aqueous 75 wt.% solution of Al(NO_3_)_3_∙9H_2_O (Sigma-Aldrich Chemie GmbH) under the same conditions as described above, calcined at 600 °C to decompose the aluminum nitrate into Al_2_O_3_, and finally sintered at 1650 °C. These samples were subjected to exactly the same handling and processing steps as the ZrO(NO_3_)_2_ infiltrated foams; consequently, they represent a more realistic reference material compared to RPCs, which were only infiltrated with sub-micrometer alumina without the additional salt solution processing.

### 2.2. Characterization

The infiltration fluids were characterized by dynamic light-scattering (DLS) using a Zetasizer Nano ZS (Malvern GmbH, Herrenberg, Germany) and the undiluted metal nitrate solutions or a 100 mg·L^−1^ dispersion of sub-micrometer alumina particles, respectively. For the calculation of the median solute species size the volume weighed size distribution was evaluated (d_50,3_ value). In order to chemically analyze the solute species being present in the zirconyl nitrate solutions, Raman spectroscopy was applied using an Alpha 300R spectrometer (Witec GmbH, Ulm, Germany) equipped with a 532 nm laser and a 1800 g·mm^−1^ grating. A drop of the respective salt solution was placed on a glass slide, the laser beam was focused slightly below the liquid surface, and a Raman spectrum with a spectral resolution of 1.3 cm^−1^ was recorded.

The viscosity of the alumina infiltration dispersion and the salt solutions was furthermore measured by rotational viscosimetry using a MCR-301 rheometer (Anton Paar GmbH, Graz, Austria) with parallel plate geometry, 50 mm plate diameter, and a gap of 0.5 mm between both plates. The flow curves were recorded without a preshear step in a shear rate range between 10 s^−1^ and 100 s^−1^ (salt solutions) or between 0.1 s^−1^ and 400 s^−1^ for the TM-DAR alumina dispersion. For each fluid, three specimens were measured in four consecutive runs each, resulting in 12 flow curves which were finally averaged.

The accurate content of crystal water being present in the ZrO(NO_3_)_2_∙*x*H_2_O used within this work as well as the optimal calcination temperature were determined by a thermogravimetric measurement of the as-received zirconyl nitrate in the temperature range between 20 °C and 1000 °C in air using a STA 449 F3 Jupiter device (Netzsch GmbH & Co. KG, Selb, Germany) and a heating rate of 10 K∙min^−1^. From the weight fraction of the remaining ZrO_2_ residue the water content was calculated.

The ZTA foam manufacturing process was monitored by recording the mass *m*_f_ and the geometric volume *V*_f_ of each individual specimen after (a) presintering at 1350 °C (starting foam), (b) alumina infiltration and subsequent drying, (c) zirconyl nitrate infiltration and subsequent drying, (d) calcination at 600 °C, and (e) final sintering at 1650 °C. The total geometric porosity was calculated from the relative density of each specimen, which is the mass *m*_f_ divided by the volume *V*_f_, in relation to the theoretical density of the ceramic material. For the ZTA samples, the theoretical density was estimated by the rule of mixture and the respective weight fractions of alumina and zirconia, as determined by powder X-ray diffraction (XRD) and Rietveld analysis. For the foams consisting of pure Al_2_O_3_ a theoretical density of 3.94 g∙cm^−3^ was used [41]. The total porosity was determined for (a) the noninfiltrated starting foams obtained after presintering at 1350 °C and (b) for the samples after alumina and zirconyl nitrate infiltration and the final sintering step at 1650 °C. The strut wall porosity of the finally sintered foams was calculated from the dry, buoyant and water-filled weight of the foams as determined by the water immersion/Archimedes’ method in adaption of the DIN EN 623-2:1993-11 standard [42]. All porosity results were averaged for all 15 specimens of each sample series.

For selected samples, the hollow strut volume was investigated by means of mercury intrusion porosimetry (MIP) with an Autopore 9500 IV device (Micromeritics, Norcross, GA, USA). A mercury filling pressure of 38 mbar was applied and raised to 82.8 bar during the course of the experiment. The cumulative mercury intrusion volume was considered and normalized to the weight of the respective specimen for a further analysis.

The phase composition of the ZTA foams was determined by powder XRD using a PANalytical Empyrean diffractometer (Almelo, The Netherlands) with Bragg–Brentano geometry and Cu Kα_1_/α_2_ radiation (PANalytical). Beforehand, the foam samples were ball-milled in an alumina grinding bowl; the obtained powder was then filled in a backloading sample holder and measured in *θ/θ* reflection geometry with a *2θ* range of 10° to 120°. The obtained diffraction data were then processed by the Rietveld technique using the TOPAS Academic 5 software package [43,44].

The compressive strength of the foam samples was determined by a TIRAtest 2825 uniaxial mechanical testing machine with 150 mm circular loading plates, which was operated with a head moving speed of 1 mm∙min^−1^ (TIRA GmbH, Schalkau, Germany). A cardboard piece of 1 mm thickness was placed between the loading plates and the foam sample in order to ensure a homogeneous load of the entire cellular structure. From the data the maximum force was extracted, normalized to the loaded geometric surface area of the sample and used for the calculation of the compressive strength. The results of all specimens from a particular sample series were evaluated using a three-parameter Weibull distribution within the Visual-XSel 14 software package (CRGRAPH GbR, Munich, Germany) [45,46]. From this distribution, the average compressive strength σ_cf_ (Weibull scale parameter) together with a minimum compressive strength σ_cf,min_ (Weibull location parameter) were deduced for the respective sample series.

The microstructure of selected specimens was characterized by scanning electron microscopy using a Scios Dualbeam microscope (FEI, Hillsboro, OR, USA) equipped with a secondary electron and backscattered electron detector. In addition, microcomputed tomography (µ-CT) was applied for the characterization of foam specimens at different processing levels as well as individual foam struts of selected samples (nanotom S tomograph, Phoenix/GE Sensing & Inspection, Wunstorf, Germany). For each measurement, a set of 1080 radiographs with a resolution of 2304 pixels × 2304 pixels was collected with an exposure time of 2 s per image. The distances between detector and X-ray source (FDD) and between sample and X-ray source (FOD) were adjusted to result in a voxel size of (24 µm)^3^ for the representation of the foam structure as a whole or of (0.625 µm)^3^ for high-resolution measurements of individual struts. Data acquisition and reconstruction was performed with the Phoenix Datos│X 2.0 software package (Phoenix/GE Sensing & Inspection). For calculations of the strut and cell size distributions, the CTAnalyser 1.18 software package was used (CTAn, Skyscan/Bruker microCT, Kontich, Belgium). The import of the collected CT data into the CTAn software and the differential thresholding-based binarization procedure preceding the actual calculations were performed as described elsewhere [40]. The cell size and strut thickness distributions were calculated after filling the hollow strut cavities by performing a morphological closing operation (erosion/dilatation) in CTAn using a round kernel with r = 6 µm, and subsequent despeckling as described in a previous work [47]. The average thickness of a strut filament as well as the dimension of the hollow strut cavities in sintered foams were determined with the CTAn software without a preceding closing operation. From these data the average coating thickness of the alumina dispersion on a PU template foam directly after manufacturing was calculated under consideration of the observed linear shrinkage during sintering, the strut wall porosity and the volumetric solid content of the coating dispersion. 

## 3. Results and Discussion

### 3.1. Starting Materials and Infiltration Fluids

According to thermogravimetry, the thermal decomposition of the ZrO(NO_3_)_2_∙*x*H_2_O starting material proceeds in three subsequent steps in a temperature interval between 100 °C and 600 °C (Figure 2). Over the course of the first two decomposition reactions with *T*_max._ = 120 °C and 210 °C, respectively, an equivalent of HNO_3_ and H_2_O each split off forming an intermediate ZrO(OH)_2_ phase in accordance with the observed weight loss. In the final decomposition step this hydroxide phase is dehydrated into a mixture of 73 wt.% monoclinic ZrO_2_ and 27 wt.% tetragonal ZrO_2_ with respect to a XRD analysis of the decomposition residue. In accord with literature data, the formation of zirconia is virtually finished at 380 °C [48]; however, a small weight loss is recorded up to a temperature of 600 °C resulting in a total oxide yield of 38 wt.%. Consequently, a temperature of 600 °C has been selected for calcination of the ZrO(NO_3_)_2_ infiltrated samples. From the weight of the ZrO_2_ residue a water content of 5.4 molecules H_2_O per mole ZrO(NO_3_)_2_ has been calculated.

The solubility of ZrO(NO_3_)_2_∙*x*H_2_O in water or aqueous HNO_3_ is sufficiently high so that solutions with a Zr^4+^ concentration of approximately 1.1 mol∙L^−1^ (equivalent to 30 wt.% of ZrO(NO_3_)_2_∙5.4H_2_O or 140 g∙L^−1^ of ZrO_2_) can be prepared. However, from the dynamic light-scattering (DLS) measurements on these ZrO(NO_3_)_2_ solutions a significant influence of the H_3_O^+^ concentration on the size of the zirconium species being present in the respective solution was found (Figure 3 and Table 2). For ZrO(NO_3_)_2_ dissolved in demineralized water an average size of the solute species of 8 nm was measured by DLS, which is in good accord to sol particles of a condensed zirconium oxygen species [49]. In contrast thereto, the mean solute diameter in solutions of ZrO(NO_3_)_2_ in 1 mol∙L^−1^ and 2 mol∙L^−1^ HNO_3_, respectively, is 1 nm, indicating a significantly lower degree of aggregation. The measured solute dimension is in good agreement to the tetrameric [Zr_4_(OH)_8_(H_2_O)_16_]^8+^ cluster cation, which is the predominant species in Zr^4+^ solutions with c(H_3_O^+^) ≥ 1 mol∙L^−1^ [50,51]. The significantly different size of the zirconium species being present in HNO_3_ and H_2_O, respectively, is expected to result in an effect on the (physical) properties of the solutions on the one hand side and on the infiltration behavior of the porous alumina substrate on the other.

The chemical structure of the zirconium solute species in HNO_3_ and H_2_O has been further investigated by vibrational spectroscopy. From Raman spectra recorded from the respective ZrO(NO_3_)_2_ solutions, the DLS results were confirmed; in solutions containing HNO_3_ the characteristic bands of the [Zr_4_(OH)_8_(H_2_O)_16_]^8+^ cluster cation were observed at 430 cm^−1^, 460 cm^−1^, and 580 cm^−1^, respectively (Figure 4), which is in good accord to literature spectra of the [Zr_4_(OH)_8_(H_2_O)_16_]^8+^ cluster [51]. In contrast thereto, these bands were absent in the zirconyl nitrate solution in demineralized water; instead, new bands were observed at 380 cm^−1^ and 630 cm^−1^, which can be assigned to a condensed ZrO_2_ sol species [51].

The alumina infiltration dispersion and the zirconyl nitrate solutions were further characterized by rotational viscosimetry. A significantly different flow behavior of the ZrO(NO_3_)_2_ solution in demineralized water compared to the acidified systems was found (Figure 5 and Table 2). By fitting the measured rheological data with the Ostwald–de Waele model (power law; η = *K*∙γ^(*n* − 1)^) a flow index of *n* = 0.75 is the result for the aqueous ZrO(NO_3_)_2_ solution indicating a non-Newtonian, shear-thinning flow behavior [52]. This is in good agreement to the flow behavior of a disperse system containing particles in the nanometer range. For the acidified solutions in 1 mol∙L^−1^ and 2 mol∙L^−1^ HNO_3_ a Newtonian flow behavior was observed with a flow index of 0.94 and 1.05, respectively, in accord to systems containing molecularly dispersed species. Furthermore, the consistency index of the ZrO(NO_3_)_2_ solutions decreases from *K* = 5.6 mPa∙s for demineralized water as solvent to 1.6 mPa∙s for the ZrO(NO_3_)_2_ solution in 2 mol∙L^−1^ HNO_3_. This indicates a significantly lower viscosity for the acidified solutions as a consequence of the lower degree of aggregation between the zirconium species.

The alumina infiltration dispersion is characterized by an almost ideal Newtonian flow behavior with a flow index *n* of 0.92 and *K* = 14.0 mPa∙s, which represents a low viscosity for a disperse system containing 31 vol.% of particulates [40]. The low viscosity of the sub-micrometer alumina dispersion is a direct consequence of the almost spherical shape of the TM-DAR powder particles. This represents an optimal perquisite for the capillary force-driven penetration into the hollow strut cavities in RPCs.

The initial alumina foams are characterized by a total geometric porosity (*V*_pores_/*V*_foam_) of 93.8 ± 0.2 vol.% after sintering at 1350 °C. The strut wall porosity (*V*_strut pores_/*V*_strut_) as determined by the Archimedes method is 40 ± 2 vol.%. The isotropic linear shrinkage is 8.2% and equivalent to a volumetric shrinkage of 22.6% in relation to the dimensions of the original PU foam template of (19.8 mm)^3^. From a morphometric analysis of µ-CT reconstruction data, an average strut diameter of 0.43 ± 0.17 mm and an average cell size of 2.6 ± 0.2 mm were determined (Table 3). The low shrinkage value indicates a partial sintering of the alumina strut material due to the low sintering temperature of 1350 °C. Consequently, the average cell size of 2.6 mm is slightly larger compared to similar 20 ppi Al_2_O_3_ foams sintered at 1650 °C with a fully densified strut material. For these, a linear shrinkage of 13.7% and an average cell size of 2.5 mm have been reported [47].

### 3.2. Hollow Strut Infiltration of Alumina Foams with Alumina and Zirconia

From a preceding work it is known that the ambient pressure infiltration of alumina RPCs with a dispersion of sub-micrometer alumina particles is effective for filling the hollow strut cavities in these foams completely [14]. This is a consequence of the low viscosity and high flowability of the sub-micrometer alumina dispersion allowing it to penetrate the hollow strut cavities through cracks and defects in the struts. The origin of these longitudinal strut cracks is the wetting behavior of the dispersion used in the foam manufacturing and the so-called “Kantenfluchtproblem” (see above) [17,18,19]. Within this work, the porosity of the starting foams has been adjusted to high values of 93.8% after sintering at 1350 °C or 91.5% after the final sintering at 1650 °C. Consequently, the coating of the template with the alumina dispersion directly after manufacturing is thin with 0.21 ± 0.16 mm, on average. After drying, an average coating thickness of 0.14 ± 0.11 mm is the result for the green RPC, which shrinks to 0.13 ± 0.10 mm (1350 °C) or 0.12 ± 0.09 mm (1650 °C) after sintering. Due to this low coating thickness on the struts of the PU template, the wetting of the strut edges is especially poor resulting in a significant amount of longitudinal strut cracks. Nevertheless, these flaws are essential for an effective loading of the hollow strut cavities with the sub-micrometer alumina dispersion. The dimension of the longitudinal strut cracks can be roughly estimated by mercury intrusion porosimetry (MIP); the Hg penetrates the hollow strut cavities through these strut defects as well; consequently, the equivalent pore size, as determined by MIP of 50 µm, correlates to the dimension of the strut defects (Figure 6). However, the drawbacks of the Washburn model usually applied for the pore size estimation must be considered in this context [53]. A more realistic approximation of the strut crack diameter is accessible by a manual measurement from µ-CT slices, which gives an average strut crack width of 25 ± 10 µm.

Within this work, the expected results for the hollow strut infiltration were observed with respect to µ-CT data; apart from occasionally occurring air bubbles, the hollow strut cavities are completely loaded with alumina. Consequently, the hollow strut infiltration with sub-micrometer alumina particles results in an average weight increase of *m*_inf_ = 0.40 ± 0.06 g, which is 27.4% with respect to the weight of the initial alumina foams. As the starting foams were the same, no significant differences exist between the individual sample series (Figure 7). The average porosity of the foams after the alumina infiltration without further sintering is 92.1 ± 0.2%. The mean porosity decrease is then 1.7 ± 0.2% and in good accord to previous studies dealing with the hollow strut infiltration of RPCs [14,25,26].

The theoretically expected weight gain by the alumina loading of the hollow strut cavities *m*_th_, was approximated by a simple geometric calculation from the volume of the hollow struts *V*_hs_, the volume fraction of alumina in the infiltration dispersion *ϕ*_ds_, and the theoretical density of alumina *ρ*_th_ according to Equation (1):*m*_th_ = *V*_hs_ ∙ *ϕ*_ds_ ∙ *ρ*_th_(1)

The theoretical density of alumina (3.94 g∙cm^−3^ as determined by Franco Jr., et al., see ref. [41]) and the alumina loading in the infiltration dispersion (0.309) are known, but the volume of the hollow struts in the initial alumina RPCs presintered at 1350 °C is ambiguous. A simple approach for estimating *V*_hs_ is by applying the volumetric shrinkage of 22.6% as observed during the sintering procedure on the strut volume of the initial PU foam template [54]. The template volume is accessible from the average weight of the 8 cm^3^ PU foams of 0.243 ± 0.007 g and a He pycnometry density of the PU material of 1.107 ± 0.001 g∙cm^−3^. Finally, resulting in a total volume of 0.170 mL for the hollow strut cavities in one alumina foam sintered at 1350 °C. Consequently, a theoretical mass increase by the hollow strut infiltration of 0.207 g is the result according to Equation (1). However, this approach is problematic as an isotropic shrinkage of the strut material in direction of the strut’s center of gravity is assumed, but the real shrinkage behavior of the hollow strut structures in RPCs is still unknown.

Therefore, the hollow strut volume *V*_hs_ of the presintered starting foams has been investigated by µ-CT measurements. This analysis is based on subtracting the material volume of a foam, including the hollow strut cavities from the same dataset for which the hollow strut cavities have been closed by an erosion/dilatation procedure within the CTAn software. This evaluation results in a hollow strut volume of *V*_hs_ = 0.103 ± 0.002 mL based on an average of five individual foam specimens, and consequently gives a value of *m*_th_ = 0.126 ± 0.002 g for the theoretical mass increase by the hollow strut infiltration. As possible error sources the presence of pores, e.g., air bubbles inside the strut material, as well as an overthickening of the struts during the erosion/dilatation procedure applied on the µ-CT data, have to be considered. Both effects would result in a potential overestimation of *V*_hs_.

With respect to the observed weight gain *m*_inf_ = 0.40 ± 0.06 g after the alumina hollow strut infiltration process this is a severe underestimation. Therefore, an additional coating of the outer strut surface, which takes place in any case, has to be discussed in this context [27]. According to µ-CT morphometric analyses of the initial foams before and after the hollow strut infiltration, a thickening of the struts by approximately 80 µm, being equivalent to an average coating thickness of 40 µm, was observed (Table 3). Nevertheless, it should be noted that this observed thickening is within the standard deviation of the mean strut thickness, but indicates at least a general trend.

The difference between the experimental and the theoretical weight gain after the hollow strut infiltration, *m*_inf_ − *m*_th_ = 0.27 g, can be applied for estimating the average thickening of the struts. In a simplified approach it is assumed that the hollow struts ingest an alumina amount equivalent to *m*_th_ and the excess material is deposited on the strut surface. For this purpose, a simple geometrical model approximating all struts of the cellular structure as one single cylindrical rod was used (rod model), which considers the strut curvature, for details see the work of Betke, et al. in refs. [39] and [47]. Eventually, an average coating thickness of 33 µm is calculated from the difference *m*_inf_ − *m*_th_ equivalent to a thickening of the strut diameter by 66 µm. A volumetric packing density of the alumina particles in the coating of 31% of the theoretical density of alumina has been assumed based on the alumina volume fraction of the initial infiltration dispersion [55,56]. However, from SEM micrographs of alumina-infiltrated foams an inhomogeneous coating with sub-micrometer Al_2_O_3_ particles with a thickness in the range of 1 to 20 µm was observed. This probably indicates that more material resides inside the struts and the loading of the hollow strut cavities with alumina exceeds the theoretical value of *m*_th_.

During the infiltration of the alumina hollow strut loaded foams with ZrO(NO_3_)_2_ solution a significant amount of the sub-micrometer alumina powder deposited on the strut surface is rinsed off of the foams as no sintering is performed after the alumina infiltration. On average, 0.28 ± 0.03 g of sub-micrometer alumina remains for one individual foam sample (Figure 7). The ZrO(NO_3_)_2_ solution is not agitated during the course of the infiltration; consequently, rinsing off the alumina coating from the foam struts instead of washing out the alumina material from the hollow struts is expected. This is in good accordance with µ-CT morphometric investigations of the alumina and ZrO(NO_3_)_2_ infiltrated foams, for which a reduction of the average strut thickness by 60 µm, being equivalent to a decrease of the average coating thickness by 30 µm, was observed with respect to the strut thickness of the alumina-only infiltrated foams (Table 3).

However, the alumina amount of 0.28 ± 0.03 g incorporated in one foam specimen is still significantly larger than the theoretical mass increase expected from the hollow strut infiltration of 0.126 ± 0.002 g. In this context, a filtration effect of the partially sintered; still, porous strut material during the hollow strut infiltration process should be discussed [57]. During the course of the infiltration, the initial foam’s porous alumina strut material can absorb some of the water from the infiltration dispersion being inside the hollow strut cavities already—in analogy to the slip casting of ceramic parts using porous plaster molds [58]. This is expected in a local increase of the alumina volume fraction of the infiltration dispersion, and would, therefore, result in a larger amount of alumina deposited inside the hollow strut cavities (Figure 6, Right). The amount of 0.28 ± 0.03 g alumina incorporated into a hollow strut volume of 0.103 ± 0.002 mL corresponds to a volumetric filling of the struts by 70 vol.%, which is close to the volume ratio in an idealized closest sphere packing of 74 vol.%. This agrees to mercury intrusion porosimetry (MIP) investigations of the foams with and without alumina infiltration finally sintered at 1650 °C. For the noninfiltrated specimen a hollow strut volume of 0.101 mL was observed, whereas a residual hollow strut volume of 0.032 mL, which is 31.7 vol.%, remains for an alumina-infiltrated foam (Figure 6, Left).

The amount of ZrO(NO_3_)_2_—and consequently the amount of ZrO_2_ after the final sintering, as deposited in the strut material of the alumina-infiltrated Al_2_O_3_ foams—increases linearly with the HNO_3_ concentration in the respective infiltration solution (Figure 7). For an aqueous solution of zirconyl nitrate a total ZrO_2_ content of 1.2 ± 0.05 wt.% was found after the sintering at 1650 °C, which increases to 1.6 ± 0.05 wt.% for a solution containing 2 mol∙L^−1^ HNO_3_ (Table 4). This is most likely a consequence of the decreasing viscosity of the zirconyl nitrate solution with increasing concentration of HNO_3_ (Figure 5). Together with the smaller size of the zirconium species in the acidified systems this results in a more effective loading of the porous strut material with the ZrO(NO_3_)_2_ solution. The ratio between monoclinic and tetragonal ZrO_2_ after the final sintering of the foams is not affected by the properties of the infiltration solutions containing only ZrO(NO_3_)_2_. For all sample series, a ratio of t-ZrO_2_:m-ZrO_2_ of ≈ 0.25 was observed according to the results of the Rietveld analyses (Figure 8, Left). For the samples which were infiltrated with a Y-doped zirconyl nitrate solution, however, significantly more tetragonal zirconia is retained after the final sintering (t-ZrO_2_:m-ZrO_2_ ≈ 6.7; Table 4). Nevertheless, it should be noted that the XRD investigations were performed on strut fragments collected after the compressive strength measurements, which were grinded in a ball mill; an interference of this mechanical load on the Al_2_O_3_/ZrO_2_ material with the ratio between m-ZrO_2_ and t-ZrO_2_ cannot be excluded.

In another sample series the ZrO_2_ concentration in the strut material was increased by repeated infiltrations with zirconyl nitrate solution. For this purpose, the ZrO(NO_3_)_2_ deposited in the strut material after the preceding infiltration cycle was calcined into ZrO_2_ at 600 °C to avoid leaching from the sample. The amount of ZrO_2_ increases linearly with the cycle number from 1.6 ± 0.05 wt.% for one cycle to 4.9 ± 0.05 wt.% for three consecutive infiltration–decomposition cycles (2 mol∙L^−1^ HNO_3_; Table 4). Again, a ratio t-ZrO_2_:m-ZrO_2_ of ≈ 0.25 was observed for all sample series according to the results of the Rietveld analyses (Figure 8, Right).

After the final sintering step at 1650 °C, an average isotropic linear shrinkage of 15.3% equivalent to a volumetric shrinkage of 39.2% in relation to the dimensions of the original PU foam template of (19.8 mm)^3^ was measured. The total porosity of all samples spans over the narrow range between 91.4 vol.% and 91.7 vol.%, without a clear correlation to the infiltration processing parameters (Table 4). Consequently, the total porosity reduction after the infiltration processing is 2.3 vol.% on average (Table 3). This porosity decrease is a result of the hollow strut infiltration and, in equal parts, the sample volume decrease after the final sintering at 1650 °C (Figure 7). The strut wall porosity (*V*_strut pores_/*V*_strut_), as determined by the Archimedes method, ranges between 18 ± 1 vol.% and 26 ± 5 vol.%, without a clear correlation to the infiltration processing and is in accord to studies of similar alumina RPCs [39]. From µ-CT data a mean strut thickness of 0.42 ± 0.14 mm and an average cell size of 2.4 ± 0.2 mm were determined for selected alumina and zirconia infiltrated samples (Table 3), which is in good agreement to results from a previous study of 20 ppi alumina foams [47].

### 3.3. Strut Microstructure of Alumina Foams after Alumina and Zirconia Infiltration

The microstructure of individual struts of selected alumina and zirconia loaded foam samples was investigated by µ-CT and backscattered electron (BSE) microscopy coupled with energy dispersive X-ray spectroscopy (EDS). In the reconstructed µ-CT data, a clear concentration of a material with higher X-ray absorption inside the struts—predominantly the former hollow strut cavities filled with sub-micrometer alumina particles—was observed (Figure 9a–d). Therefore, these regions contain more of the higher absorbing ZrO_2_ phase, which indicates a preferred infiltration with ZrO(NO_3_)_2_ solution of the hollow strut cavities loaded with sub-micrometer alumina particles. This is a direct consequence of the very different microstructures of the already presintered strut material and the loose packing of unsintered alumina particles inside the hollow strut cavities. The latter contains a significantly higher amount of voids due to the incomplete filling of the struts, which can be filled with the ZrO(NO_3_)_2_ solution (see sketch in Figure 9e). The significantly higher amount of zirconia inside the formerly hollow strut cavities was also observed in BSE micrographs and EDS spectra collected on the respective strut regions (Figure 10).

In addition, both the BSE micrographs of strut cross-sections and the µ-CT data show the mending of the typical defects in RPCs—namely the longitudinal strut cracks originating from an improper coating of the PU template with the ceramic dispersion on the sharp edges of the PU foam’s struts (Figure 9f,g). Consequently, these intrinsically weak regions inside the RPC structure are selectively mended by the hollow strut infiltration with sub-micrometer alumina particles and reinforced by the preferred loading of these spots with a ZrO_2_ transformation toughening phase.

With respect to the strut microstructure the formation of cracks inside the material incorporated into the formerly hollow strut cavities was observed by µ-CT (Figure 9d). This is a result of the incomplete loading of the hollow struts with alumina–zirconia of approximately 70 vol.% (30 vol.% were not loaded) and the volumetric shrinkage during the final sintering step at 1650 °C. This shrinkage is higher for the unsintered Al_2_O_3_/ZrO_2_ material inside the struts compared to the already presintered outer strut material resulting in the as observed formation of cracks. Nevertheless, these cracks run within the Al_2_O_3_/ZrO_2_ material inside the struts preferentially, and not along the interface between the alumina outer strut regions and the alumina–zirconia material inside the struts. This indicates a sufficient bonding between the reinforcement phase inside the foam struts and the existing structure of the (initially) hollow struts.

### 3.4. Mechanical Behavior of Alumina Foams after Alumina and Zirconia Infiltration

The effect of the hollow strut infiltration on the compressive strength is clearly visible with an average strength of 0.6 MPa for the sample series without infiltration, 1.0 MPa for samples with sub-micrometer alumina hollow strut infiltration (results of Scheffler et al., see ref. [14]) and 1.5 MPa for the foams with alumina and zirconia hollow strut infiltration (Table 4). For the reference sample series, which was consecutively infiltrated with sub-micrometer alumina and aluminum nitrate solution, the average strength is 1.1 MPa and very similar to specimens which were loaded with alumina only.

The strength improvement after the alumina hollow strut infiltration is by a factor of 2, on average, for samples with comparable total porosity, which is in good accord to previous studies [14,25,26]. If an additional ZrO_2_ transformation toughening phase is added into the strut material, the compressive strength increases by another 50%, or by a factor of 3, on average, with respect to specimens without hollow strut infiltration. However, the total porosity of all RPC specimens within this work ranges between 91.4% and 91.8%, which has to be considered in the further interpretation of the mechanical strength results. With respect to the collapse mechanism a brittle crushing is observed during the compression experiments of all sample series and is also identified in SEM images collected on the fragments of the respective tested specimens.

Thus, the obtained data were modeled with the Gibson–Ashby (GA) relation describing the crushing behavior of brittle, cellular materials (Equation (2)) for the evaluation of the compressive strength of the differently infiltrated alumina RPCs [59]. The GA model was applied in its logarithmic expression (Equation (3)), allowing a simple linear regression analysis. This allows to separate the effect of the porosity on the compressive strength from the actual impact of (micro)structural improvements, like hollow strut infiltration or a transformation toughening reinforcement, see Equations (2) and (3).

*σ*_cf_ = *C*_6_ ∙ (*ρ*_rel_)*^n^* ∙ *σ*_fs_(2)

ln(*σ*_cf_) = ln(*C*_6_) + *n* ∙ ln(*ρ*_rel_) *+* ln(*σ*_fs_)(3)

According to Equation (2), the compressive strength *σ*_cf_ of a brittle, cellular sample is a function of its relative density and the bending strength *σ*_fs_ of the bulk strut material. The density exponent *n* describes the effect of the porosity on the mechanical strength; in the original GA model *n* is 1.5. However, *n* is correlated to the concentration of defects within the strut material. For brittle, open-celled ceramic or glass foams, *n* typically ranges between 1.5 and 3 [60,61,62]. The constant *C*_6_ is related to the cellular geometry of the foam; for cellular ceramics made by the polymer sponge replication technique, usually a value of 0.65 is used for *C*_6_ [59]. A fixed value of 400 MPa has been applied for the bending strength *σ*_fs_ of the bulk strut material for the alumina-only RPCs (noninfiltrated and alumina-infiltrated) [4]. The constant *C*_6_ has been set to 0.65 according to the original work of Gibson and Ashby [59] and the exponent *n* was refined as the only free variable.

For both, the samples without hollow strut infiltration, and the RPCs, which were infiltrated with alumina only, a reasonable GA fit with equation 3 is the result. The GA exponent *n* converged to 2.5 for the material model of noninfiltrated RPCs, whereas *n* is reduced to 2.25 for the GA model of alumina-only infiltrated foams (Figure 11, Left). This suggests an improvement of the cellular structure, namely the mending of RPC-specific defects within the ceramic struts as described previously. For the GA model of the alumina–zirconia infiltrated RPCs, the fit was altered slightly: As no obvious change of the strut microstructure with respect to the occurrence of defects, like cracks or pores, was observed for the Al_2_O_3_/ZrO_2_-loaded samples in comparison to the alumina-only RPCs, the GA exponent *n* has been fixed to 2.25 and the bending strength of the bulk strut material *σ*_fs_ has been refined as free variable.

Finally, the strength data obtained for the different sample series with alumina–zirconia infiltration were successfully modeled using equation 3 with *n* = 2.25, *C*_6_ = 0.65, and a bending strength of the strut material *σ*_fs_ converged to 575 MPa (Figure 11, left). The obtained bending strength is in good accord to literature data for bulk ZTA ceramics made by shaping and sintering at ambient pressure [63,64]. This represents the expected toughening effect of the strut material due to the zirconia loading.

Equation (3) has been also applied for normalizing the strength results of all sample series to a total porosity of 91.6%, being equivalent to a relative density of 0.082 (dashed line in Figure 11). This is the numeric average between the porosity of the samples from this study and the data collected on alumina-only infiltrated foams by Scheffler et al., see ref. [14]. After this porosity normalization, the impact of the processing parameters “HNO_3_ concentration in the ZrO(NO_3_)_2_ solution” and “number of consecutive infiltration cycles” was analyzed decoupled from the slight porosity variations between the individual sample series.

Interestingly, no significant correlation between the conditions and the cycle number of the ZrO(NO_3_)_2_ infiltration and the average compressive strength *σ*_cf_ of the respective sample series normalized to a porosity of 91.6% was observed (Figure 11, Right). The *σ*_cf_ values fluctuate between 1.36 MPa and 1.60 Mpa, with an average of 1.47 MPa for all Al_2_O_3_/ZrO_2_ infiltrated samples. This represents an increase between 30% and 52%, or 40% on average, compared to the Al_2_O_3_/Al(NO_3_)_3_ infiltrated reference material, which has been subjected to exactly the same processing steps. The porosity-corrected strength increase with respect to noninfiltrated foams (*σ*_cf_ = 0.48 MPa at 91.6% porosity) is 206%, on average. This clearly indicates the effectiveness of the hollow strut infiltration, in general, and of refitting an additional ZrO_2_ transformation toughening phase into the strut material, in particular.

The missing correlation between the ZrO_2_ deposition conditions and the final normalized compressive strength probably indicates that small amounts of ZrO_2_ loaded into the intrinsically weak regions of RPCs are already sufficient for an effective transformation toughening mechanism. This becomes apparent in the virtually equal strength of the samples infiltrated once with aqueous ZrO(NO_3_)_2_ solution (1.2 wt.% ZrO_2_; *σ*_fc_ = 1.46 MPa) and the foams infiltrated three times with a solution of zirconyl nitrate in 2 mol∙L^−1^ HNO_3_ (4.9 wt.% ZrO_2_; *σ*_fc_ = 1.51 MPa).

The doping of the ZrO_2_ phase generated from the ZrO(NO_3_)_2_ material with Y does not show an effect on the compressive strength of the final RPCs, despite the significant effect on the phase evolution of the ZrO_2_ material (see above). The observed compressive strength is in line with the results for foams treated with pure zirconyl nitrate; the reason for this effect is, however, unclear up to now. 

Nevertheless, more interesting is the course of the minimum compressive strength parameter σ_cf,min_, as determined by the three-parameter Weibull analysis of the respective strength data. For the sample series infiltrated with ZrO(NO_3_)_2_ in H_2_O and in 1 mol∙L^−1^ and 2 mol∙L^−1^ HNO_3_, respectively, a linear increase of the minimum compressive strength from 0.79 MPa to 1.14 MPa was observed (Table 4). This is in line with the amount of ZrO_2_ deposited in the struts of the respective RPCs, which follows the same linear trend. Consequently, the infiltration with acidified zirconyl nitrate solution resulting in an increased ZrO_2_ loading does not inevitably improve the average strength, but significantly improves the strength σ_cf,min_ to which the RPCs can be loaded without any failure. Nevertheless, for the samples infiltrated 1×, 2×, and 3× with ZrO(NO_3_)_2_ σ_cf,min_ varies between 0.96 MPa and 1.45 MPa without a clear correlation to the infiltration cycle number. However, for these samples, it must be considered that the multiple infiltration procedure significantly increases the number of processing steps; each handling step of a RPC is a potential extra source of structural damage to the cellular material. Thus, the expected strength increase due to the increased amount of ZrO_2_ for the multiply infiltrated RPCs is probably foiled by these structural defects related to the additional processing steps.

The positive effect of the zirconia loading into alumina hollow strut-infiltrated alumina RPCs is clearly visible, especially if the results are evaluated against the strength of the reference foams infiltrated with alumina and aluminum nitrate solution. For these, all processing steps were identical to the samples treated with alumina and zirconyl nitrate solutions; nevertheless, the alumina–zirconia infiltrated RPCs possess a significantly higher compressive strength.

The origin of this strength increase is most likely the transformation toughening mechanism in ZTA ceramics, which is known for a long time [34,35]. The stress-induced phase transformation from tetragonal to monoclinic zirconia involving a significant volume increase of the ZrO_2_ grains is of central importance in this context. The crack expansion is hindered by both, absorption of the activation energy for the *t*/*m* transformation of ZrO_2_, and the compressive stress exerted on the alumina matrix by the expanding ZrO_2_ grains [36].

Nevertheless, other phenomena, specifically those related to the significantly different coefficients of thermal expansion (CTE), 6.3 × 10^−6^ K^−1^ for alumina [65] and 8.0 × 10^−6^ K^−1^ for pure monoclinic zirconia [66], may be of significance: Some part of the sub-micrometer alumina particles was collected on the strut surface; this porous coating layer is loaded with ZrO(NO_3_)_2_ in the same manner like the material incorporated into the hollow struts. Thus, the outer strut regions contain more zirconia than the original strut material (see Figure 9). On cooling of the samples after the sintering procedure, these strut regions contract to a slightly higher extent compared to the inner strut material due to the higher concentration of zirconia. Consequently, an intrinsic residual compressive stress is generated on the strut, which has already been proven as a possible reinforcement for mullite/alumina RPCs [67]. This mechanism is most effective for a fully closed coating around the strut possessing a higher CTE compared to the inner strut material. Nevertheless, according to analyses of the strut microstructure of the alumina–zirconia infiltrated foams, this is not the case for the samples investigated within this work. Consequently, CTE-related effects might play a role locally, but the transformation toughening inside the strut material is most likely the dominant reason for the significant increase in compressive strength. In summary, additional experiments are necessary for a deeper analysis of these effects.

Interestingly, the porosity-corrected strength of the ZTA foams made by the alumina hollow strut infiltration, with subsequent refitting of the ZrO_2_ transformation toughening phase via ZrO(NO_3_)_2_, is superior to ZTA foams, which were hollow strut-infiltrated with a dispersion of ZTA powder directly [26,28]. For these RPCs the normalized compressive strength is in line with pure, hollow strut-infiltrated alumina foams (Figure 12). This is astonishing, as the samples described by Vogt et al. and Chen et al., see refs. [26,28], contain a significantly higher amount of the zirconia transformation toughening phase (16 wt.% to 30 wt.%). Most likely, the reason for the high mechanical strength of the ZTA foams described in the above mentioned study is the inhomogeneous distribution of the ZrO_2_ phase in the ceramic struts. Due to the microstructural characteristics of the alumina hollow strut-infiltrated foams within this work, the ZrO_2_ transformation toughening phase is installed selectively in these regions which contain the previously incorporated sub-micrometer alumina particles. Ideally, these are the intrinsically weak spots of ceramic foams made by the replica technique—namely, the hollow struts and, especially, the longitudinal cracks within the strut material. Consequently, the ZrO_2_ is accumulated in exactly these regions resulting in a significant toughening effect even at very low zirconia concentrations ranging between 1 wt.% and 5 wt.% with respect to the entire strut material.

In summary, the combination of the already established hollow strut infiltration of RPCs with ceramic dispersions and the capillary force driven loading of residual porosity within the ceramic struts with a precursor for the generation of a transformation toughening phase is very effective. The amount of material consumed (sub-micrometer alumina dispersion and ZrO(NO_3_)_2_ solution) is very low for an individual foam and the processing fluids can be easily reused for other samples. Finally, ceramic foams with a compressive strength of 1.5 MPa at a high porosity of almost 92% were obtained, which is almost a threefold strength increase compared to conventional alumina RPCs at the same porosity level. In addition, the cell geometry as well as the outer strut shape, which determine the fluid transportation properties within this kind of cellular structures, is not affected, as the functionalization takes place inside the hollow struts exclusively.

## 4. Conclusions

Within the present work a functionalization approach for reticulated porous ceramics (RPCs) manufactured by the polymer sponge replication technique has been investigated. This functionalization process exploited the intrinsic hollow strut cavities in RPCs, which were infiltrated with a dispersion of sub-micrometer alumina particles, at first. In a second infiltration step with an aqueous ZrO(NO_3_)_2_ solution, the residual strut porosity was loaded with a ZrO_2_ precursor compound. After a final sintering, Al_2_O_3_/ZrO_2_ (ZTA) composite foams with almost dense struts, but a complex strut microstructure with respect to the ZrO_2_ distribution, were obtained.

Due to the selective mending of the structural defects typically found in RPCs, the hollow strut infiltration resulted in a significant increase of the compressive strength of the functionalized foams. For alumina-only infiltrated samples the porosity-normalized compressive strength increased by 119%, on average, with respect to the noninfiltrated specimens; refitting of a zirconia transformation toughening phase into the strut material resulted in an increase of 206% with respect to the initial RPCs without infiltration. The ZrO_2_ transformation toughening phase accumulated in the formerly hollow strut areas as well as longitudinal cracks within the strut material, which were previously filled with sub-micrometer alumina particles. Consequently, the transformation toughening is most effective in these intrinsically weak spots of RPCs.

Besides the investigation of the hollow strut infiltration with alumina and ZrO(NO_3_)_2_ in general, the properties of the infiltration fluids were investigated as a function of the respective solvent acidity. An increasing HNO_3_ concentration in the ZrO(NO_3_)_2_ solutions resulted in a significant decrease in the dimensions of the respective Zr solute species. This was corroborated by a shift of the rheological behavior from a non-Newtonian, shear-thinning flow in aqueous solutions of ZrO(NO_3_)_2_ to an almost ideal Newtonian behavior for solutions of ZrO(NO_3_)_2_ in nitric acid. The lower viscosity and smaller Zr solute species of the acidified zirconyl nitrate solutions resulted in an increase of the ZrO_2_ fraction within the strut material after infiltration and calcination.

Interestingly, the average mechanical strength of the ZTA foams was virtually independent from the characteristics of the ZrO(NO_3_)_2_ solutions and the number of consecutive infiltration cycles, as well. Nevertheless, an increasing minimum compressive strength with increasing acidity of the ZrO(NO_3_)_2_ solution was detected.

In conclusion, the hollow strut functionalization is a very effective way of increasing the mechanical strength of RPCs at a minimal loss of porosity. The refitting of a zirconia transformation toughening into the struts selectively reinforces the neuralgic flaws in RPC structures; a significant improve of the compressive strength with minimal material effort is the result. The manufacturing of cellular ceramics with a compressive strength exceeding 1.5 MPa at an open porosity of ≥ 91.5 vol.% becomes feasible. Future challenges are mostly related to a simplification of the comparatively complex handling procedures involving several consecutive processing steps.

## Figures and Tables

**Figure 1 materials-12-01886-f001:**
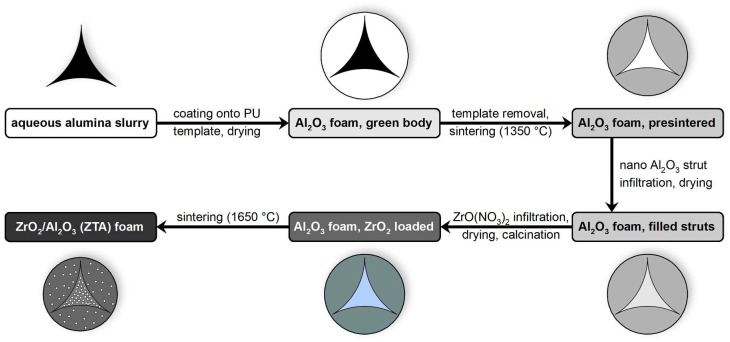
Process flow chart for the manufacturing of zirconia-toughened alumina (ZTA) foams with dense struts by hollow strut infiltration with alumina, and subsequent infiltration of the microstructural porosity with ZrO(NO_3_)_2_ solution.

**Figure 2 materials-12-01886-f002:**
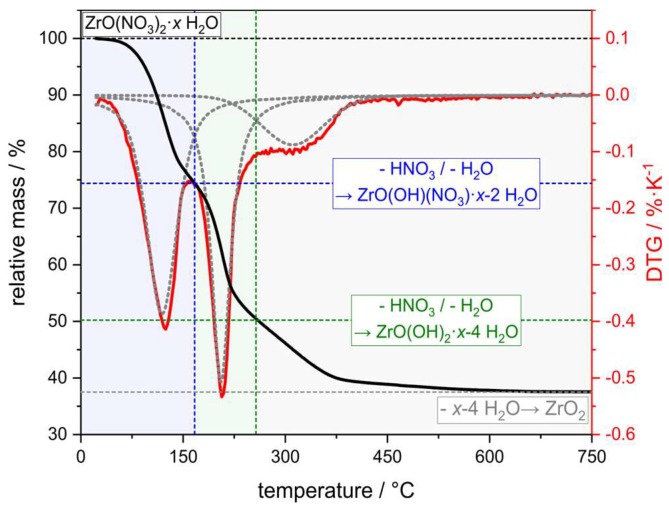
Thermogravimetric (TG) analysis of the thermal decomposition of the ZrO(NO_3_)_2_∙*x*H_2_O salt used within this work (black line: TG curve; red line: first derivative of the TG data, DTG). From the weight fraction of the residual ZrO_2_, a water content per formula unit of *x* = 5.4 has been calculated.

**Figure 3 materials-12-01886-f003:**
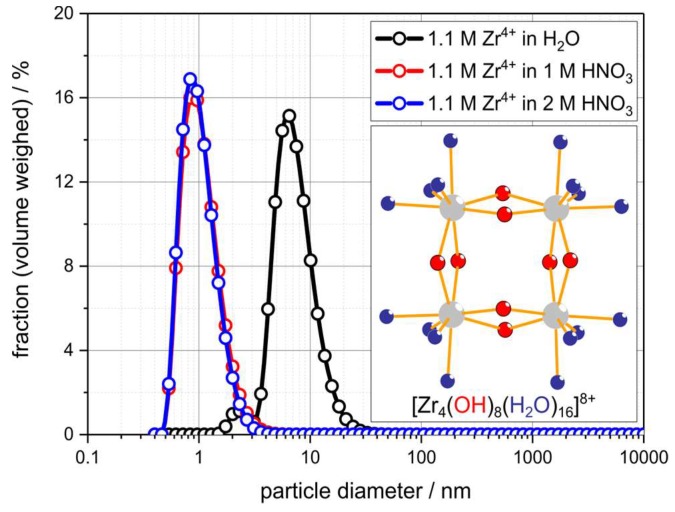
Dynamic light-scattering results for ZrO(NO_3_)_2_ solutions in demineralized water and nitric acid with c = 1 mol∙L^−1^ and 2 mol∙L^−1^. Inset: Molecular structure of the tetrameric [Zr_4_(OH)_8_(H_2_O)_16_]^8+^ cluster ion, which is the predominant species in the solutions containing HNO_3_.

**Figure 4 materials-12-01886-f004:**
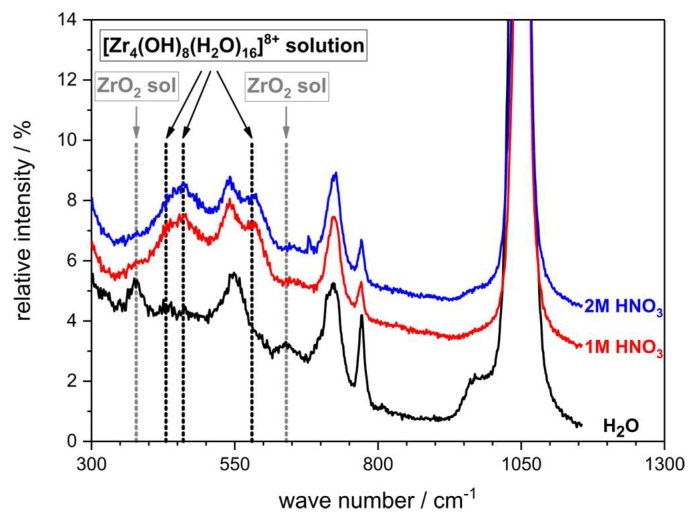
Raman spectra of ZrO(NO_3_)_2_ solutions in demineralized water and nitric acid with c = 1 mol∙L^−1^ and 2 mol∙L^−1^. In the acidified solutions, Raman bands characteristic for the tetrameric [Zr_4_(OH)_8_(H_2_O)_16_]^8+^ cluster ion are found. The signals above 650 cm^−1^ can be assigned to the vibrations inside the nitrate anion.

**Figure 5 materials-12-01886-f005:**
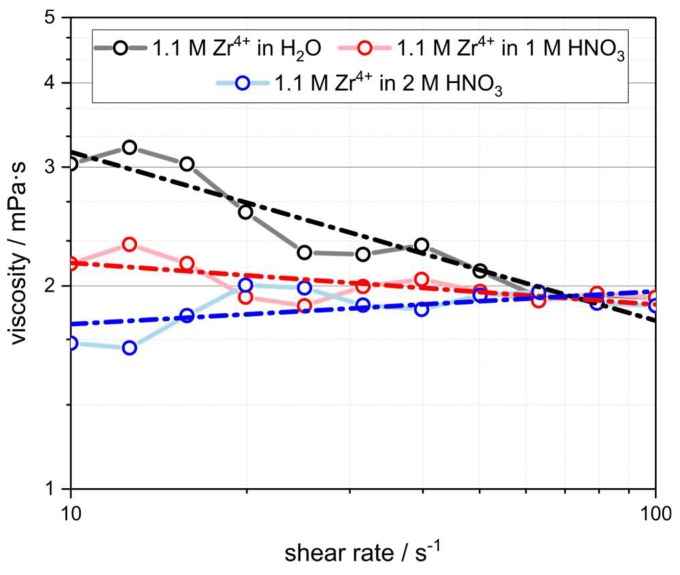
Rotational viscosimetry of ZrO(NO_3_)_2_ solutions in demineralized water and nitric acid with c = 1 mol∙L^−1^ and 2 mol∙L^−1^. Dashed lines: Corresponding fits of the rheological data with the Ostwald–de Waele (power law) model.

**Figure 6 materials-12-01886-f006:**
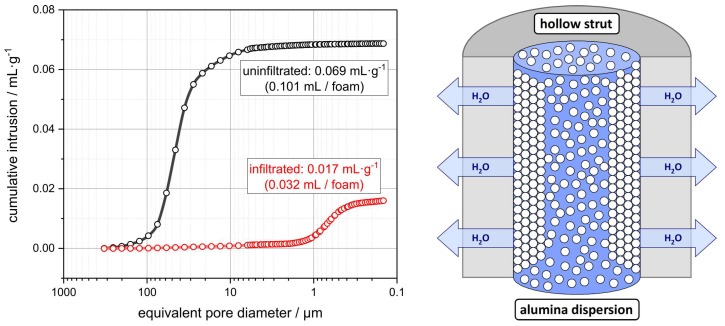
(**Left**) Plot of the cumulative intrusion volume from MIP experiments for cellular alumina without hollow strut infiltration (black) and after filling of the hollow strut cavities with submicron alumina particles (red). Both foams were subjected to a final sintering step at 1650 °C. The cumulative volume of the hollow struts is reduced by 68.3 vol.% after the infiltration process. (**Right**) Schematic representation of the filtration of the alumina dispersion through the porous strut material during the hollow strut infiltration resulting in a volumetric loading of the hollow strut cavity significantly exceeding the alumina load in the infiltration dispersion.

**Figure 7 materials-12-01886-f007:**
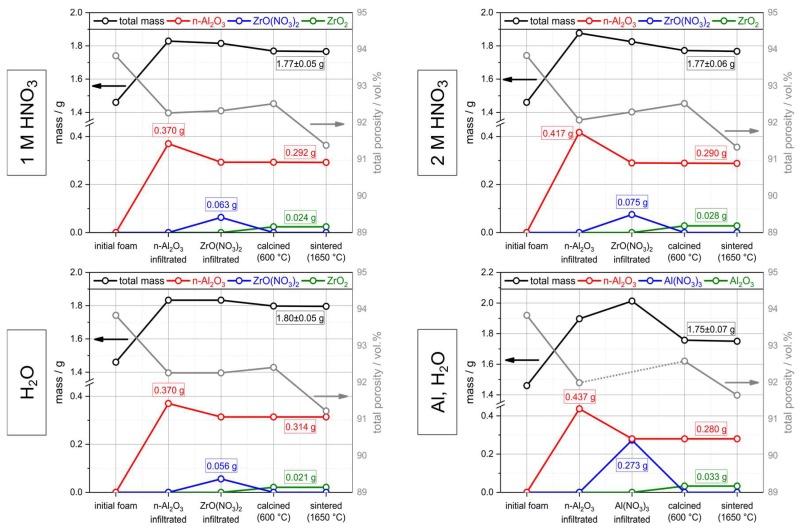
Mass and porosity evolution of selected alumina foam series (averaged from 15 specimens each) during the hollow strut loading with sub-micrometer alumina particles and the subsequent infiltration with ZrO(NO_3_)_2_ or Al(NO_3_)_3_ solutions, respectively, in demineralized water and nitric acid with c = 1 mol∙L^−1^ and 2 mol∙L^−1^. The weight fraction of zirconia after sintering has been determined by Rietveld analysis and the weight fraction of ZrO(NO_3_)_2_ after the salt infiltration has been calculated therefrom by consideration of an oxide yield of 38 wt.% during the calcination step. The amount of Al(NO_3_)_3_ for the Al–nitrate-infiltrated foams has been estimated by assuming a loading of the hollow strut cavities with 0.28 g submicron alumina (average for all sample series) for one foam.

**Figure 8 materials-12-01886-f008:**
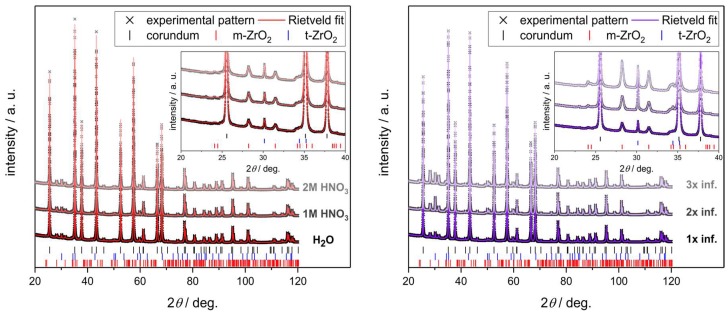
Powder X-ray diffraction (XRD) patterns with the corresponding Rietveld fits for different Al_2_O_3_-ZrO_2_ foams. (**Left**) Variation of the HNO_3_ concentration in the ZrO(NO_3_)_2_ solution; (**Right**) variation of the infiltration cycle numbers (ZrO(NO_3_)_2_ in 2 mol∙L^−1^ nitric acid). The inset shows the most prominent reflections of the monoclinic and tetragonal ZrO_2_ phases.

**Figure 9 materials-12-01886-f009:**
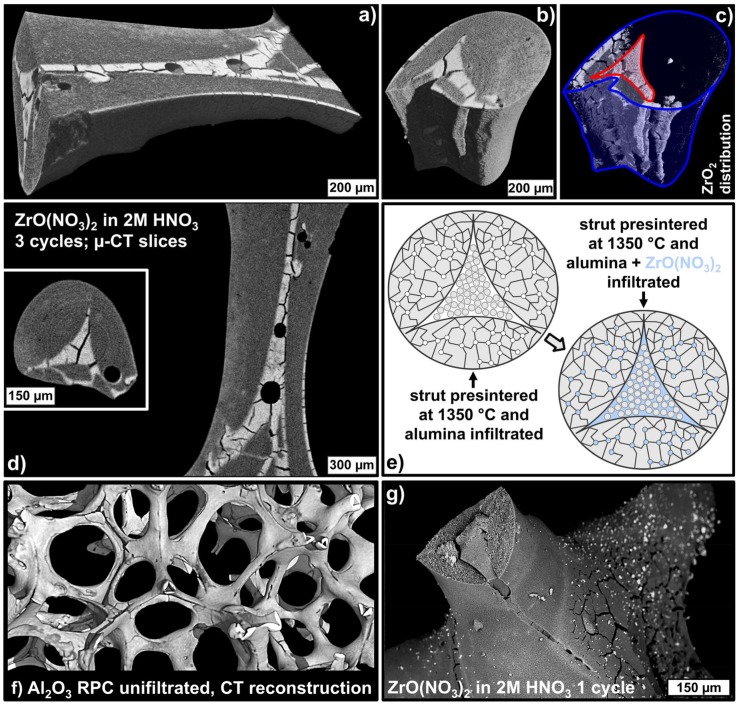
Micro-CT analysis of the microstructure of a single strut from an alumina foam infiltrated with a sub-micrometer alumina dispersion and subsequently three times with a solution of ZrO(NO_3_)_2_ in 2 mol∙L^−1^ nitric acid (**a**–**d**). The specimen was sintered at 1650 °C. Zirconia concentrates in the former hollow strut regions (**c**), which are filled by a loose packing of alumina particles after the infiltration with sub-micrometer Al_2_O_3_ (**e**). Structure of the foam struts before (**f**) and after the alumina–zirconia infiltration processing (**g**) showing the characteristic strut defects and their mending after the hollow strut infiltration.

**Figure 10 materials-12-01886-f010:**
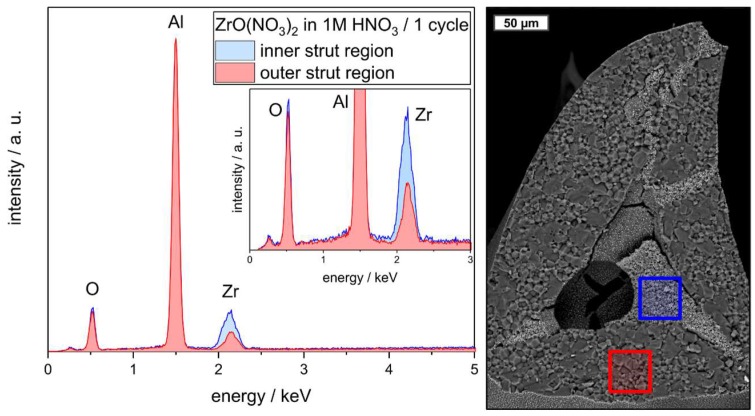
(**Right**) Backscattered electron micrograph of a strut cross-section of an Al_2_O_3_–ZrO_2_ foam (ZrO(NO_3_)_2_ in 1 mol∙L^−1^ nitric acid; one infiltration cycle). (**Left**) Energy dispersive X-ray (EDS) spectra collected on an area inside the formerly hollow strut (blue) and on the strut material of the initial foam (red).

**Figure 11 materials-12-01886-f011:**
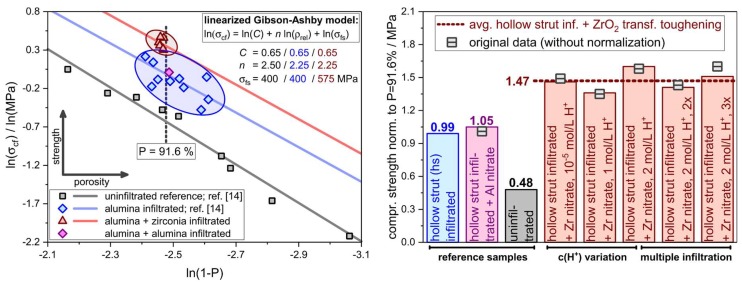
(**Left**) Compressive strength data of 20 ppi Al_2_O_3_ foams without hollow strut infiltration (gray squares), with alumina infiltration (blue diamonds), with alumina plus ZrO(NO_3_)_2_ (→ ZrO_2_) infiltration (red triangles), and with alumina plus Al(NO_3_)_3_ (→ Al_2_O_3_) infiltration (reference sample, purple diamond); each symbol corresponds to a complete sample series with 15 specimens. The regression lines show the corresponding fits with the Gibson–Ashby model for the strength ↔ porosity correlation, the inset shows the derived values for the parameters C and n as well as the bending strength σ_fs_ of the respective strut material. The data for noninfiltrated alumina foams (gray squares) and samples with alumina hollow strut infiltration (blue diamonds) were adapted from Scheffler et al., see ref. [14]. (**Right**) compressive strength data normalized to a porosity of 91.6% (column chart); the numbers on top of each column represent the average strength of the noninfiltrated, alumina-infiltrated, alumina plus ZrO(NO_3_)_2_ (→ ZrO_2_) infiltrated and alumina plus Al(NO_3_)_3_ (→ Al_2_O_3_) sample series.

**Figure 12 materials-12-01886-f012:**
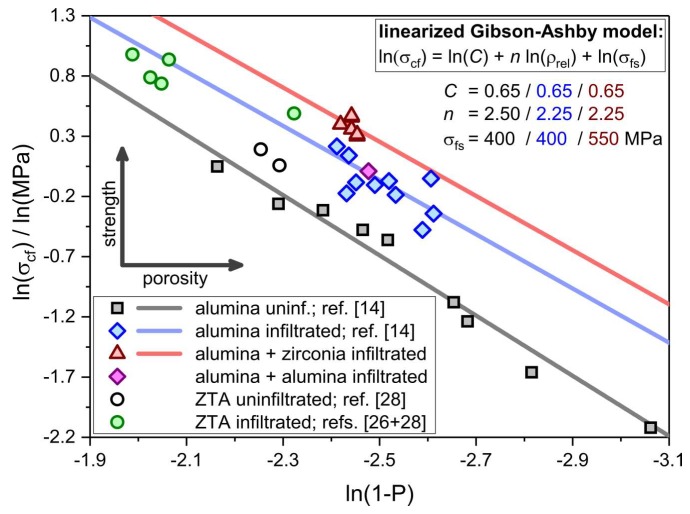
Compressive strength data of ZTA foams made by the zirconyl nitrate infiltration approach (red triangles; this work) and of ZTA foams without (black circles; adapted from Chen et al., see ref. [28]) and with hollow strut infiltration with ZTA particles (green circles; adapted from Vogt et al. and Chen et al., see refs. [26,28]).

**Table 1 materials-12-01886-t001:** Composition of the alumina dispersions used for the foam manufacturing and the infiltration of the hollow strut cavities.

Component	Material	Dispersion Foam Manufacturing	Dispersion Strut Infiltration
Alumina powder	CT 3000 SG, d_50_ = 0.5 µm ^a^	78.3 wt.%	-/-
	TM-DAR, d_50_ = 0.1 µm ^a^	-/-	63.8 wt.%
Dispersant	demineralized water	19.6 wt.%	35.5 wt.%
Deflocculant	ethanolammonium citrate (Dolapix CE64)	0.8 wt.%	0.6 wt.%
Binder	polyvinylalcohol (Optapix PA 4G)	1.2 wt.%	-/-
Defoamer	polyalkylene glycolether (Contraspum K1012)	0.1 wt.%	0.1 wt.%

^a^ according to the manufacturers data sheet.

**Table 2 materials-12-01886-t002:** Properties of the alumina dispersion used for the hollow strut infiltration and the infiltration solutions made of ZrO(NO_3_)_2_∙5.4H_2_O (M = 328.4 g∙mol^−1^) and Al(NO_3_)_3_∙9H_2_O (M = 375.0 g∙mol^−1^).

Fluid	Solvent/wt.%	Metal Salt/wt.%	Density/g∙cm^−3^	c(M^n+^)/mol∙L^−1^	OxideContent /g∙L^−1^	d_50,3_ ^a^ /nm	K ^b^ /mPa∙s
TM-DAR H_2_O	36	64	-/-	-/-	638	236 ± 88	14.0 ± 1.0
ZrO(NO_3_)_2_ H_2_O	70	30	1.19 ± 0.02	1.09 ± 0.02	134.0 ± 1.6	7.8 ± 3.6	5.6 ± 0.5
ZrO(NO_3_)_2_ 1M HNO_3_	70	30	1.23 ± 0.02	1.13 ± 0.02	138.8 ± 1.8	1.1 ± 0.5	2.5 ± 0.2
ZrO(NO_3_)_2_ 2M HNO_3_	70	30	1.26 ± 0.02	1.15 ± 0.02	141.2 ± 1.7	1.1 ± 0.4	1.6 ± 0.1
Al(NO_3_)_3_ H_2_O	25	75	1.43 ± 0.02	1.91 ± 0.03	97.2 ± 2.4	1.1 ± 0.3	18.2 ± 0.2

^a^ median size of solute species, volume weighed; ^b^ consistency index according to the Ostwald–de Waele law (equivalent to the viscosity at a shear rate of 1 s^−1^).

**Table 3 materials-12-01886-t003:** Total porosity and morphometric data of alumina foams at different processing steps.

Processing Level	Geometric Porosity ^a^/%	Strut Thickness (CT) ^b^/mm	Cell Size (CT) ^b^/mm
initial foam (1350 °C)	93.8 ± 0.2	0.43 ± 0.17	2.6 ± 0.2
alumina-infiltrated	92.1 ± 0.2	0.51 ± 0.13	2.6 ± 0.2
alumina and zirconia infiltrated	92.5 ± 0.2	0.45 ± 0.14	2.5 ± 0.2
sintered foam (1650 °C)	91.5 ± 0.2	0.42 ± 0.14	2.4 ± 0.2

^a^ average of all sample series; ^b^ µ-CT investigations of selected samples

**Table 4 materials-12-01886-t004:** Total porosity prior to and after the alumina–zirconia infiltration processing of alumina RPCs (average of 15 foams per sample series) together with the amount of ZrO_2_ loaded into the respective samples and the compressive strength data. The parameters σ_cf_ and σ_cf,min_ correspond to the average and minimum compressive strength according to a three-parameter Weibull analysis of the strength results. The sample designator corresponds to the type of infiltration solution (Zr: ZrO(NO_3_)_2_; Al: Al(NO_3_)_3_), the solvent used, and the number of consecutive infiltrations.

Sample	Infiltration	Initial Porosity ^a^/%	Final Porosity ^b^/%	ZrO_2_/wt.% (m:t)	σ_cf_/MPa	σ_cf,min_/MPa
Zr_H_2_O	Al_2_O_3_/ZrO(NO_3_)_2_ in demin. H_2_O	93.8 ± 0.2	91.4 ± 0.3	1.2 (79:21)	1.49 ± 0.36	0.79
Zr_1M_HNO_3_	Al_2_O_3_/ZrO(NO_3_)_2_ in 1M HNO_3_	93.8 ± 0.2	91.5 ± 0.4	1.4 (80:20)	1.35 ± 0.21	0.97
Zr_2M_HNO_3_	Al_2_O_3_/ZrO(NO_3_)_2_ in 2M HNO_3_	93.8 ± 0.2	91.5 ± 0.4	1.6 (79:21)	1.58 ± 0.24	1.14
Zr_2M_HNO_3__2x	Al_2_O_3_/ZrO(NO_3_)_2_ in 2M HNO_3_ (2x)	93.8 ± 0.2	91.4 ± 0.4	3.3 (71:29)	1.43 ± 0.29	0.96
Zr_2M_HNO_3__3x	Al_2_O_3_/ZrO(NO_3_)_2_ in 2M HNO_3_ (3x)	94.0 ± 0.1	91.5 ± 0.3	4.9 (84:16)	1.60 ± 0.07	1.45
Zr_Y_2M_HNO_3_	Al_2_O_3_/ZrO(NO_3_)_2_ + Y in 2M HNO_3_	93.7 ± 0.1	91.5 ± 0.4	1.4 (13:87)	1.37 ± 0.26	0.87
Al_H_2_O_ref.	Al_2_O_3_/Al(NO_3_)_3_ in H_2_O	93.8 ± 0.2	91.7 ± 0.5	-/-	1.01 ± 0.28	0.43
uninf._ref. ^c^	-/-	-/-	91.8 ± 0.3	-/-	0.56 ± 0.21	0.24

^a^ after first sintering of the noninfiltrated foams at 1350 °C; ^b^ after the final sintering of the sub-micrometer Al_2_O_3_ and ZrO(NO_3_)_2_ infiltrated foams at 1650 °C; ^c^ literature data from Scheffler et al., see ref. [14].

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
