# Peer review of "Refitting of Zirconia Toughening into Open-Cellular Alumina Foams by Infiltration with Zirconyl Nitrate"

_materials, 2019, doi:10.3390/ma12121886_

Round 1
Reviewer 1 Report
The authors have submitted a paper dealing with reinforcement of alumina foams via foam infiltration with an alumina slurry and zirconyl nitrate solutions. The investigation was carefully performed and described in details. However, the paper is rather long and becomes wordy in some paragraphs. Moreover, the authors present a lot of speculative hypotheses. It is then difficult to follow the main message of the paper. This reviewer recommends to shorten the paper. There are also some particular items given below that need to be taken into account during the paper revision:
1. The fine alumina suspension had to enter the hollow struts through large openings and cracks. Identify and document these openings. What was the cause of these defects? It seems that the alumina infiltration repaired a bad coating of the polymer sponge.
2. Rheological behaviour measured at a shear rate range from 10 to 100 s-1 doesn’t represent the behaviour during infiltration; the real shear rate must have been lower.
3. What was the porosity of ceramics in the strut walls after presintering and after sintering? Does the infiltration influence the porosity of the sintered wall ceramics?
4. What was the reason to apply the three-parameter Weibull distribution instead of two-parameter one? Show the Weibull plot and check the confidence intervals for the Weibull parameters.
Miscellaneous:
- Check and correct the legend to Table 4
- Check subscript for compressive strength in Fig. 11
Author Response
Please find the responses to the points of reviewer 1 in the attached .pdf file.

Reviewer 2 Report
In the present work, the concept of the hollow strut infiltration in alumina reticulated porous ceramics is combined with the transformation toughening through ZrO2 grains within the Al2O3 matrix. The manuscript is interesting and the overall structure of the paper is good. However, the following changes / improvements are recommended:
- English language and style are fine/minor spell check required.
- Some recent developments published in the Materials journal should be considered, showing a continuity between the present work and those reported in the literature on similar topics.
- In this paper the authors discuss about porous materials such as open-cell ceramic foams. Accordingly, a brief presentation of new-developed syntactic foams (see Fiedler et al, Linul et al., Movahedi et al. etc.) would add value to the present work. Please consider this.
- The conclusions section should be written more concise (too many details are presented).
- Some stress-strain curves would be very useful to see the behavior of the investigated foams.
- How was initial/final porosity measured? How was foam compressive strength determined? etc. The standard after which the tests were performed should be presented.
- Collapse mechanisms of investigated foams would see very well on some SEM images (obtained from the tested samples).
Author Response
Please find the responses to the points of reviewer 2 in the attached .pdf file.

Round 2
Reviewer 1 Report
The authors have addressed all reviewer’s comments and revised the paper accordingly. There is only one comment remaining. The authors are highly advised to add SEM images of the foam with the strut cracks and openings before infiltration as well as images of the final foam after infiltration.
Author Response
An image showing the strut microstructure, namely the longitudinal cracks along the struts and the hollow strut cavities, before the infiltration has been included into the manuscript (figure 9f). In addition, a micrograph showing the strut microstructure after alumina-zirconia infiltration and representing the mending of the above mentioned flaws has been added as well (figure 9g).